# Developing shared understanding of pre-eclampsia in Haiti and Zimbabwe using Theory of Change

**Tanya Robbins**[1]*, **Mickias Musiyiwa**[2], **Muchabayiwa Francis Gidiri**[3], **Violet Mambo**[3], **Carwyn Hill**[4], **Jane Sandall**[1], **Charlotte Hanlon**[5,6], **Andrew H. Shennan**[1]

**1** Department of Women & Children's Health, School of Life Course and Population Science, King's College London, St Thomas' Hospital, London, United Kingdom, **2** Department of History, Heritage and Knowledge Systems, Faculty of Arts and Humanities, University of Zimbabwe, Harare, Zimbabwe, **3** Department of Obstetrics and Gynaecology, College of Health Sciences, University of Zimbabwe, Harare, Zimbabwe, **4** Hope Health Action, Cap-Haitien, Haiti, **5** Centre for Global Mental Health, Health Service and Population Research Department, and WHO Collaborating Centre for Mental Health Research and Training, Institute of Psychiatry, Psychology and Neuroscience, King's College London, London, United Kingdom, **6** Department of Psychiatry, WHO Collaborating Centre for Mental Health Research and Capacity-Building, School of Medicine, College of Health Sciences, Addis Ababa University, Addis Ababa, Ethiopia

* tanya.robbins@kcl.ac.uk

**Data Availability Statement:** The data generated and analysed during this study are uploaded as Supporting Information.

## Abstract

Pre-eclampsia, a complex and multi-system disorder specific to pregnancy, is a leading cause of preventable maternal and perinatal deaths in low-resource settings. Early detection and appropriate intervention with management of hypertension, prevention of eclampsia and timely delivery are effective at reducing mortality and morbidity. Outcomes can be greatly improved with the provision and uptake of good quality care. Cultural contexts of maternal care, social practices and expectations around pregnancy and childbirth profoundly shape understanding and prioritisation when it comes to seeking out care. Few studies have addressed health education specifically targeting pre-eclampsia in low resource settings. The existing literature has limited descriptions of contextual barriers to care or of the intervention development processes employed. More engaging, holistic approaches to pre-eclampsia education for women and families that recognise the challenges they face and that support a shared understanding of the disorder, are needed. We describe our experience of developing a Theory of Change (ToC) as part of the co-production of educational resources for pre-eclampsia in Haiti and Zimbabwe.

## Introduction

Pre-eclampsia, a complex and multi-system disorder specific to pregnancy, is a leading cause of preventable maternal and perinatal deaths in low resource settings [1–3]. Early detection and appropriate intervention with management of hypertension, prevention of eclampsia and timely delivery are effective at reducing mortality and morbidity. Outcomes can be greatly improved with the provision and uptake of good quality care [4, 5]. In many low resource

**Funding:** This work was co-funded by The Medical Research Council and Arts and Humanities Research Council (MC_PC_MR/R024510/1). TR, AS, JS and CH are funded through the ASSET research programme, supported by the UK's National Institute of Health Research (NIHR) (NIHR Global Health Research Unit on Health Systems Strengthening in Sub-Saharan Africa at King's College London (16/136/54)) using UK aid from the UK Government. CH also receives support from NIHR through grant NIHR200842 from AMARI as part of the DELTAS Africa Initiative [DEL- [15-01]. JS is an NIHR Senior Investigator and is supported by the National Institute for Health Research (NIHR) Applied Research Collaboration South London (NIHR ARC South London) at King's College Hospital NHS Foundation Trust. The views expressed in this publication are those of the authors and not necessarily those of the NHS, the National Institute for Health Research or the Department of Health and Social Care, England. The funders had no role in study design, data collection and analysis, decision to publish, or preparation of the manuscript.

**Competing interests:** The authors have declared that no competing interests exist.

settings, political and economic fragility and resultant weak health systems leave women struggling to access inadequate, poor quality services. Nevertheless, when financial and structural obstacles are removed, uptake and acceptance of care is not universal [6–8]. Studies of local perceptions of pre-eclampsia and eclampsia have reported limited biomedical knowledge [9–12]. Cultural contexts of maternal care, social practices and expectations around pregnancy and childbirth profoundly shape understanding and prioritisation when it comes to seeking out care. Seizures during pregnancy are often understood to be caused by spiritual, emotional or social problems and may not be associated with hypertension or pregnancy. Care seeking is linked to local interpretations of causes of the problems observed. Women suffering from pre-eclampsia may be taken to faith or traditional healers as a first resort and health facilities only when critically unwell, if at all. Even when conceptualised as an illness related to pregnancy and requiring biomedical care, awareness of symptoms and signs of pre-eclampsia among women and their families is poor across different settings [13–15]. However, the communication of biomedical information about risks has been shown to be insufficient to induce behaviour change.

Moreover, quality of care and lack of timely escalation in health facilities exacerbate community level barriers to care for pre-eclampsia. Scarce resources, outdated guidelines and insufficient support systems challenge providers struggling to care for women and babies [16–18]. Social hierarchies and punitive cultures within health facilities also limit communication with women and quality of care. While efforts to improve quality of care to drive improvements in utilisation of services are needed, maternal care-seeking and decision-making are situated within broader social, cultural and structural contexts [19].

Few studies have been published on health education specifically targeting pre-eclampsia in low resource settings. The existing literature offers limited descriptions of contextual barriers to care or of the intervention development processes employed. A pictorial card displaying six symptoms associated with pre-eclampsia was developed and piloted in Jamaica (upper middle-income country). Mothers were surveyed before and after distribution of the cards and interviews were conducted with health professionals. The authors reported a statistically significant improvement in mothers' ability to recognise symptoms that should be acted upon as well as improving the knowledge of health workers [20]. A mobile phone educational application used in Iran (UMIC) was shown to improve women's knowledge of signs and symptoms of pre-eclampsia [21]. However, there was no weblink to the app or inclusion of a detailed summary of its content published. Other studies have focused on the psychological impacts of pre-eclampsia and health education in the postnatal period [22].

Health promotion, education or behaviour change interventions often involve multiple components with interacting proposed mechanisms of action. Theory-informed design and evaluation of interventions aims to understand how, why and in what context they may be effective [23]. Theory of change (ToC) maps can be used to illustrate these relationships. Transparent reporting on the practical application of ToC can improve understanding of the benefits of its use [24].

Maternal health inequities that exist across and within countries reflect wider social disparities and determinants of health [25]. Political and social drivers influence the burden of disease in different contexts [26–28]. A lack of trust in authorities and disconnects between communities and health services has led to a lack of access or uptake of care, particularly in more vulnerable populations [29]. Community engagement aims to address contextual challenges of care. Failure to develop relationships with and listen to local communities and leaders can lead to mistrust, misinformation and marginalisation.

We identified a need for more engaging, holistic approaches to pre-eclampsia education for women and families that recognise the contextual challenges they face and that support a

shared understanding of the disorder. The HAPPEE (Humanities and Arts in Preventing Pre-eclampsia complications through community Engagement and Education) Partnership Project set out to develop culturally relevant, context specific educational resources through interdisciplinary collaboration and community engagement. An earlier phase of our project involved formative research from a qualitative study in Haiti and Zimbabwe, to explore local understanding, lived experiences and contexts of pre-eclampsia care in both settings [30]. Building on those findings, we describe our experience of developing a ToC to support the co-production of educational resources for pre-eclampsia in Haiti and Zimbabwe.

## Methods

### Components of Theory of Change

In this context we define ToC as a participatory framework used to plan and design educational interventions and their implementation. The process was iterative and developed over a number of workshops involving stakeholders from community, facility and governmental level. Following the ToC methodology, we used a backwards mapping approach, beginning with the planned impact or overall goal, linking this to long-term, intermediate and early outcomes with the processes required to move between them. These processes or the rationale for them, involved either existing evidence, formative research collected in an earlier phase of the project or local expertise and experience. Assumptions or conditions beyond the scope of the project which were required for the intended outcomes to be achieved were made explicit. We also mapped out existing initiatives, both within the formal and informal health sectors. This included existing health promotion strategies and activities. ToC encourages articulation of complexities, compared to other theory-driven approaches that focus on more linear, rigid perspectives.

### Context

Despite similarly high maternal mortality ratios in Haiti (480 per 100,000 live births) and Zimbabwe (458 per 100,000 live births), the number of births attended by skilled health staff differ significantly at 42% and 86% respectively [31–33]. This may suggest that quality of care at facilities is a more significant barrier in Zimbabwe, compared to Haiti where 'first delays' or accessing appropriate maternal care may be the greater challenge. However, our formative research has shown that educational barriers exist across settings both at community and facility level.

The HAPPEE Partnership Project was conducted in Cap Haitien, Haiti and Tafara, Mabvuku and Makumbe, close to Harare, Zimbabwe (Fig 1).

### Developing a Theory of Change

We convened four ToC workshops preceded by an initial planning workshop where we familiarised ourselves with the approach. We used a practical guideline to support facilitation [34]. Stakeholders were identified through our local partners and included community members, health professionals, representatives from non-governmental organisations (NGOs) working in the area and representatives from the Haitian and Zimbabwean ministries of health (Table 1). In each setting we held a community and organisational level workshop. These were facilitated jointly by local project co-ordinators and the UK based co-ordinator and were conducted in local languages (Haitian Creole, Shona) with translators where needed.

Workshops began with a facilitator outlining the aims of the meeting, the ToC approach and establishing ground rules to encourage contributions from all participants. We endeavoured to create a receptive and open forum within which everyone felt listened to and respected. The

**Fig 1. Project overview.**

introductory comments explicitly stated that one of the aims of the workshop was to hear a range of opinions, ideas and experiences, and the importance of listening and contributing. We then presented key findings from the formative research to inform discussions about potential outcomes. We visually built maps using sticky notes on a wall during the discussions and then later refined these, combining community and organisational level (Fig 2).

## Inclusivity in global research

Additional information regarding the ethical, cultural, and scientific considerations specific to inclusivity in global research is included in S1 Text.

## Ethical approvals

This work was a component of the HAPPEE (Humanities and Arts in Preventing Pre-eclampsia complications through community Engagement and Education) Partnership Project, approved by the Biomedical & Health Sciences, Dentistry, Medicine and Natural & Mathematical Sciences Research Ethics Subcommittee, King's College London (LRS-17/18-5526). This and local ethical approvals were in place prior to the start of the study. Participants in the semi-structured interviews and focus group discussions in phase one gave written informed consent and workshops participants provided verbal informed consent. This work was carried out with local collaborators in Haiti and Zimbabwe.

## Results

### Co-creation of films informed by Theory of Change

Following our ToC workshops, we held a partnership building meeting to develop consensus around key messages and their rationale for inclusion in the educational resources developed (Table 2). We brought together partners involved in the phase one qualitative study and participants from the organisational level ToC workshops. The key messages were grounded in the

**Table 1. Summary of Theory of Change workshops.**

| Location (level) | Participants (role/ profession) | Length of workshop | Scope of workshop | Outputs/key discussion points |
|---|---|---|---|---|
| UK | • Professor of Obstetrics <br>• Professor of Women's Health <br>• Junior Obstetrician <br>• CEO of NGO working on maternal health in Haiti <br>• Lecturer in Global Implementation science <br>• DFID (Department for International Development) representative <br>• Research Associate with experience of Human Centred Design | 2 hours | Familiarisation with approach | • Theory of Change map (S1 Fig). <br>• Consideration of rationale for education focusing on pre-eclampsia versus wider maternal health. <br>• Discussion of existing evidence supporting community-based interventions e.g. Participatory Learning and Action (PLA) and how resources developed may be used alongside this. <br>• Importance of considering the continuum of care including antenatal, intrapartum, postpartum and neonatal periods |
| Haiti (community) | • Postnatal woman <br>• Husband <br>• Traditional Birth Attendant <br>• Traditional Healer <br>• Community Health Worker <br>• Religious Leader <br>• Religious Leader | 2.5 hours | • Feedback findings from qualitative work in phase one of project <br>• Introduce to ToC approach <br>• Involve users in design <br>• Develop further understanding of context including cultural sensitivities and practices | • Theory of Change map (Fig 2). <br>• Language and meanings of different words used to describe symptoms/signs of pre-eclampsia. <br>• Discussion of different approaches to engage audiences including use of humour, stories. <br>• Consideration of how to tackle sensitive issues including family dynamics, religious and traditional practices |
| Haiti (organisational/ facility) | • Regional Director from Ministry of Health (MSPP) <br>• Hospital Director/ Consultant Paediatrician <br>• Consultant Obstetrician <br>• Labour Ward Lead Midwife <br>• Lead Community Nurse <br>• Local Project Coordinator <br>• Communication Behaviour Change Consultant <br>• CEO of Partner NGO <br>• Representative from NGO working on Health System Strengthening <br>• Paediatric Nurse and Researcher | 1.5 hours | • Feedback findings from qualitative work in phase one of project <br>• Introduce to ToC approach <br>• Involve users in design <br>• Develop further understanding of context including facility level barriers to care and wider health systems factors | • Theory of Change map (Fig 2). <br>• Using prior experience in community engagement and health education from previous campaigns e.g cholera, TB. <br>• Discussion about the best methods of reaching people considering contextual constraints—lack of electricity, mobile phone coverage. <br>• Importance of encouraging hospital as the first option for care, using community health agents. Need to check language as developing scripts/materials |
| Zimbabwe (community) | • Postnatal woman <br>• Antenatal woman <br>• Husband <br>• Husband of deceased woman <br>• Mother of deceased woman <br>• Traditional Healer <br>• Faith Healer <br>• Religious Leader (2) <br>• Traditional Birth Attendant (2) <br>• Leader of Women's Group | 3 hours | • Feedback findings from qualitative work in phase one of project <br>• Introduce to ToC approach <br>• Involve users in design <br>• Develop further understanding of context including cultural sensitivities and practices | • Theory of Change map (S2 Fig). <br>• Discussion of ideas about how to build community's trust in health facilities and providers. <br>• Consideration of how to engage specific ultra-conservative Apostolic religious sects and interdenominational engagement, importance of inclusivity. <br>• Use of real narratives to increase relatability and resonance—'talking heads' |

*(Continued)*

**Table 1.** (Continued)

| Location (level) | Participants (role/profession) | Length of workshop | Scope of workshop | Outputs/key discussion points |
|---|---|---|---|---|
| Zimbabwe (organisational/facility) | • Representatives from Ministry of Health, Family Health Department (2)<br>• Representative from Ministry of Health, Nursing Services<br>• UNFPA Programme Specialist<br>• District Medical Officer<br>• Senior Nurse<br>• Midwife<br>• Matron, City Health Department<br>• National Coordinator, White Ribbon Alliance (WRA)<br>• Junior Obstetrician (2)<br>• Consultant Obstetrician<br>• Research Midwife<br>• Behaviour Change Specialist, Community Health Intervention Research NGO<br>• Geo-Information Scientist<br>• Professor of Theatre Arts, Heritage and Knowledge Systems | 2 hours | • Feedback findings from qualitative work in phase one of project<br>• Introduce to ToC approach<br>• Involve users in design<br>• Develop further understanding of context including facility level barriers to care and wider health systems factors | • Theory of Change map (S3 Fig).<br>• Discussion about ongoing synergistic work e.g. Self-care project WRA, community dialogue on gender based violence, respectful maternity care training.<br>• Consideration of building on existing work, integrating into district/national strategies to improve sustainability.<br>• Identification of the Apostolic Women's Empowerment Trust (AWET), an organisation working with particularly vulnerably women facing barriers to accessing care in pregnancy.<br>• Careful balance between empowering through education and disempowering through didactic messaging.<br>• Reached consensus about key messages to include and groups to target including religious leaders, alternative healers, men and older female decision-makers. |

findings of our earlier qualitative work and the existing literature relating to barriers to care for pre-eclampsia. We set out to consider the crucial issues that were achievable areas to target through this approach. We reflected on how to achieve a balance between the purity of messaging versus resonance with the intended audience. There was consideration of whether a focus on simple messaging would promote clarity and the benefits of this versus acknowledging uncertainty or including more nuanced information. An example of this would be the inclusion of the message focusing on universality 'All pregnant women all over the world are at risk of pre-eclampsia' rather than highlighting particular risk factors for developing pre-eclampsia. Various mediums through which to develop shared understandings of pre-eclampsia were considered including radio, television, social media and community theatre. Our approach centred on trying to avoid paternalism and didactic methods commonly associated with health education in many settings. We aimed to value knowledge grounded in lived experience and cultural, historical and structural contexts whilst developing opportunities for shared understanding. We considered film as a channel to reflect communities' experiences back to them through compelling and immersive storytelling using real-life testimonies, drama or a combination of both. We discussed the use of visual storytelling versus dialogue in film, the depiction of conflict and challenges and how turning points in the narrative could be used to help audiences imagine alternative outcomes for women and their families. The use of humour, tragedy and local songs and proverbs were discussed as methods to engage audiences. The advantages of film in overcoming literacy barriers were balanced against the implementation challenges of screenings in resource poor settings. Our ToC approach also set out to identify potentially useful implementation strategies.

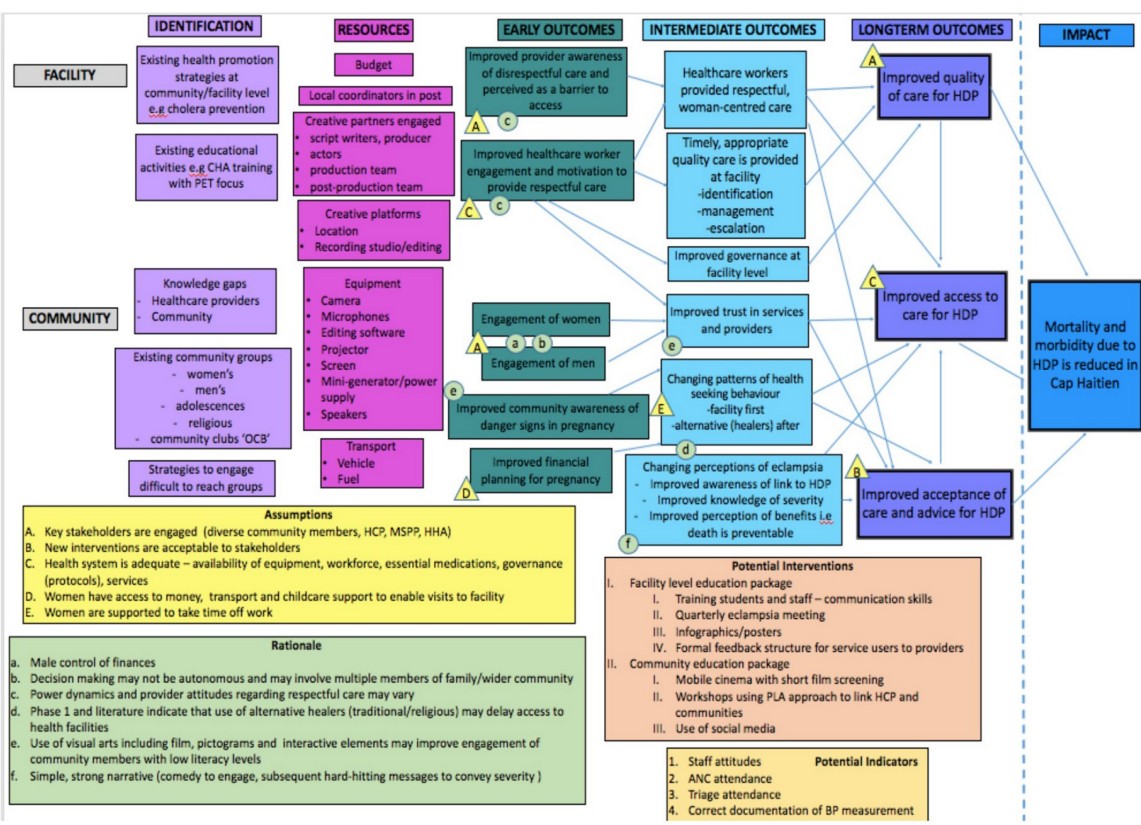

**Fig 2. Example consensus ToC map from Haiti.**

**Table 2. Consensus key messages.**

| Themes | Messages | Rationale |
|---|---|---|
| • Universality | All pregnant women all over the world are at risk of pre-eclampsia | • Importance of routine screening of blood pressure (and urine) for all pregnant women<br>• Early detection, escalation and appropriate management<br>• Addressing delays in timely detection |
| • Moving away from the culpability of women<br>• Blame and shame | Pre-eclampsia is caused by being pregnant and not what you do | • Perceived causes of symptoms and complications related to pre-eclampsia |
| • Insidious nature of pre-eclampsia<br>• Delays | Pre-eclampsia can be asymptomatic | • Importance of routine screening of blood pressure (and urine) for all pregnant women<br>• Early detection, escalation and appropriate management.<br>• Addressing delays in timely detection |
| • Empowerment | Women should be empowered and supported by their families and communities to access routine antenatal care | • Need for shared understanding and collective action |
| • Changing patterns of resort<br>• Spiritual influences | Delays can increase risks to pregnant women and their babies | • Pluralistic care seeking is common in many settings |
| • Recognising social hierarchies and influences<br>• Power dynamics | Involving men, older women and religious leaders is key to empowering women and reducing preventable deaths | • Diverse decision-makers influence maternal care seeking |
| • Shifting from fatalism to empowerment | Risks can be reduced with simple measures | • Addressing current understanding of causes<br>• Complications from pre-eclampsia are preventable where robust health systems support the provision of quality of care at facilities |

During initial meetings with our creative partners, we set out the consensus key messages and gave examples of poignant quotes from the qualitative data to contextualise them. The film development process was iterative and included script development workshops, storyboarding, engagement of local partners and identifying community actors and locations. The skill mix and approaches varied across contexts. The film development team coordinating the Haitian resources were London-based but collaborated closely with in-country partners. In both settings local-community members with no professional experience acted in the dramas.

In Haiti, a London-based team drafted a script based on the qualitative data, ToC maps and consensus key messages. This was further developed following interactive workshops in Haiti. Participants in the script development workshops included a woman of reproductive age and partner, a community leader and local healthcare workers. Discussion centred on the portrayal of the protagonists, the barriers faced to accessing care for pre-eclampsia, how they were surmounted and how realistic the representation of power dynamics in the narrative felt. Nuanced details of language were amended to improve clarity. A scene by scene read through with feedback from participants at each stage allowed us to identify areas for improvement. Throughout rehearsals this iterative process of feedback and adjustments continued. Collaboration with the community actors was key to considering issues such as the depiction of a younger, unmarried couple, family power dynamics and spiritual influences on maternal care.

In Zimbabwe, the resource development process was predominantly locally led. Regular online meetings were held with team members in London to share learning. Following the ToC workshops, the team decided that including real-life testimonies would be a powerful way to engage audiences and increase credibility and relevance. They used a combination of documentary style interviews including people with lived experience, and dramatizations acted by community actors and theatre arts students. The drama scenes depicted key themes identified in the qualitative phase, including challenges affecting decision-making processes and pluralistic care seeking for women affected by pre-eclampsia.

Following filming and editing, pilot screenings were held in Haiti and Zimbabwe in both community spaces and healthcare facilities. The films are openly available on YouTube and have been distributed locally via USB storage devices to health centres, churches, youth groups and shared on social media (S2 Text).

## Reflections

**What worked well?.** Using ToC offered many opportunities for a truly collaborative approach to educational resource development. Feedback from participants at different stages of the process was predominantly positive and included a sense of ownership, feeling listened to and enthusiasm for the benefits of connecting communities and healthcare workers based in facilities. The resulting films developed through this process received approval from communities and healthcare workers in both Haiti and Zimbabwe. Audiences reported that the characters and narratives in the films were an accurate representation of their lived experiences. Lively discussions followed many of the pilot screenings, as audience members debated the role of men in maternal care, facility level barriers in their local settings and the causes of problems during pregnancy. Some participants reported a lack of any knowledge of pre-eclampsia prior to viewing the films, some shared their own experiences of being affected by seizures in pregnancy and many audience members requested the films to be shared more widely and asked about future screenings.

**What were the challenges faced?.** We aimed to move beyond purely communicating biomedical information about pre-eclampsia, its symptoms and complications but rather

attempted to address wider contextual factors influencing help-seeking. Developing shared understanding of pre-eclampsia in settings where interpretations of problems during pregnancy are deeply rooted in sociocultural contexts, requires a sensitive and nuanced approach. Considering the range of needs and motivations of different participants was critical to bringing them together and understanding their influences on the overall objectives of the project. Our approach aimed to empower pregnant women and keep them as a central focus while recognising the importance of kinship structures and shared decision-making during pregnancy and childbirth. Fostering trust and developing shared objectives with particularly vulnerable groups can be a lengthy process. Where possible we collaborated with existing organisations embedded within communities that were aiming to strengthen links between formal and informal health systems such as the Apostolic Women's Empowerment Trust (AWET) in Zimbabwe. While essential to the work, seeking to understand historical, political and social contexts and their relationships to health is time-consuming and requires an interdisciplinary approach not always supported by current research funding mechanisms.

A lack of shared language or commonly used terms equivalent to pre-eclampsia in local languages challenged shared understanding and interpretations of illness. Various descriptive or interpretative terms related to the phenomena to which they were attributed in different settings were used, such as frothing at the mouth, acts of witchcraft or bad spirits. There were also varying levels of literacy and formal education among our participants and collaborators. We acknowledge that whilst we endeavoured to carefully facilitate workshops to ensure all voices were heard and respected, entrenched power dynamics and social hierarchies persist. Promoting equitable partnerships and meaningful collaborations is dependent on developing trust, ensuring transparency and takes time. Our ToC approach aimed to promote a bidirectional exchange of experiences and valuing different type of knowledge.

Other challenges included concerns from some participants including health care professionals and community members that providing information to women about their high blood pressure may induce fear or worsen anxiety thereby exacerbating the perceived cause of their problems. These encounters were negotiated with sensitive discussion and a recapitulation of our overall aims including improving awareness and empowering pregnant women. This collaborative approach focused on included pregnant women and their experiences, which for some included a sense of powerlessness at not understanding the nature or the cause of their condition. Pre-eclampsia can be unpredictable and variable in how and when it presents. This complexity challenges coherence in understanding across different contexts.

## Lessons learnt

- ToC can be used as a framework to support effective collaboration between women, wider community members and healthcare professionals including those based in the community. Sensitive facilitation of workshops can create safe spaces for participants to voice their ideas, renegotiate power dynamics, broaden perspectives and ensure vulnerable voices are heard.

- Theory informed approaches to intervention development encourage careful consideration of how, why and in what contexts they may work. Interventions target different levels from individual to populations or health systems. Behaviour is inherently linked to health but also to social and cultural systems. The use of frameworks that include delineation of context strengthen links between the planned activities, predicted outcomes and overall impact.

- Concurrent efforts to support improvements in quality of care at health facility level and health system strengthening are needed alongside meaningful community engagement to reduce preventable deaths due to pre-eclampsia.

## Conclusions

Community engagement is a core component of equitable, person-centred care. Involving women and the wider community in the design of innovations to improve care can build trust, transparency and ensure efforts are truly responsive to local needs and adapted to context. Creative processes and the arts can be used to empower communities, using narrative as a catalyst to stimulate meaningful collective reflections and alternative options for action and shared understanding. Community based activities in maternal health should give a voice to the most vulnerable and disadvantaged, shift power dynamics and bring together women, communities and healthcare providers.

## Supporting information

**S1 Text. PLOS questionnaire inclusivity in global health.**
(DOCX)

**S2 Text. Links to films on YouTube.**
(DOCX)

**S1 Fig. UK Theory of Change map.**
(TIFF)

**S2 Fig. Community level Theory of Change map Zimbabwe.**
(TIFF)

**S3 Fig. Organisational level Theory of Change map Zimbabwe.**
(TIFF)

## Acknowledgments

We gratefully thank all participants for their time and contributions to the study. We wish to thank Tom Besley, Bart Layton, the team at Raw, Sophie Maule, Grace Greene, Adeline Vixama, Tonderai Makaniwa, Innocent Mwapangira, Cleopas Ndaveni, Brighton Chida and Enesia Ziki.

## Author Contributions

**Conceptualization:** Tanya Robbins, Mickias Musiyiwa, Muchabayiwa Francis Gidiri, Carwyn Hill, Jane Sandall, Andrew H. Shennan.

**Data curation:** Tanya Robbins, Mickias Musiyiwa, Muchabayiwa Francis Gidiri, Violet Mambo, Andrew H. Shennan.

**Formal analysis:** Tanya Robbins, Mickias Musiyiwa, Muchabayiwa Francis Gidiri, Andrew H. Shennan.

**Funding acquisition:** Tanya Robbins, Mickias Musiyiwa, Muchabayiwa Francis Gidiri, Jane Sandall, Andrew H. Shennan.

**Investigation:** Tanya Robbins, Mickias Musiyiwa, Muchabayiwa Francis Gidiri, Violet Mambo, Carwyn Hill, Andrew H. Shennan.

**Methodology:** Tanya Robbins, Mickias Musiyiwa, Muchabayiwa Francis Gidiri, Charlotte Hanlon, Andrew H. Shennan.

**Project administration:** Tanya Robbins, Violet Mambo.

**Resources:** Tanya Robbins.

**Supervision:** Jane Sandall, Charlotte Hanlon, Andrew H. Shennan.

**Writing – original draft:** Tanya Robbins.

**Writing – review & editing:** Tanya Robbins, Mickias Musiyiwa, Muchabayiwa Francis Gidiri, Violet Mambo, Carwyn Hill, Jane Sandall, Charlotte Hanlon, Andrew H. Shennan.

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
