## [Decision Letter · Decision Letter 0]

20 Jul 2022

PGPH-D-22-00425

Developing shared understanding of pre-eclampsia in Haiti and Zimbabwe using Theory of Change

Dear Dr. Robbins,

Thank you for submitting your manuscript to PLOS Global Public Health. After careful consideration, we feel that it has merit but does not fully meet PLOS Global Public Health’s publication criteria as it currently stands. Therefore, we invite you to submit a revised version of the manuscript that addresses the points raised during the review process.

Please note that we have only been able to secure a single reviewer to assess your manuscript. We are issuing a decision on your manuscript at this point to prevent further delays in the evaluation of your manuscript. Please be aware that the editor who handles your revised manuscript might find it necessary to invite additional reviewers to assess this work once the revised manuscript is submitted. However, we will aim to proceed on the basis of this single review if possible. 

We look forward to receiving your revised manuscript.

Kind regards,

Julia Robinson

Staff Editor

Journal Requirements:

1. Please provided the full name of the IRB or ethics committee.

2. Please include a complete copy of PLOS’ questionnaire on inclusivity in global research in your revised manuscript. Our policy for research in this area aims to improve transparency in the reporting of research performed outside of researchers’ own country or community. The policy applies to researchers who have travelled to a different country to conduct research, research with Indigenous populations or their lands, and research on cultural artefacts. The questionnaire can also be requested at the journal’s discretion for any other submissions, even if these conditions are not met.  Please find more information on the policy and a link to download a blank copy of the questionnaire here: https://journals.plos.org/globalpublichealth/s/best-practices-in-research-reporting. Please upload a completed version of your questionnaire as Supporting Information when you resubmit your manuscript.

3. Please amend your detailed Financial Disclosure statement. This is published with the article. It must therefore be completed in full sentences and contain the exact wording you wish to be published.

a. State what role the funders took in the study. If the funders had no role in your study, please state: “The funders had no role in study design, data collection and analysis, decision to publish, or preparation of the manuscript.”

4. Please update the Funding Information to matched with Financial Disclosure Statement.

5.In the online submission form, you indicated that "The data generated and analysed during this study are not publicly available to protect individual anonymity but may be available from the corresponding author on reasonable request.". All PLOS journals now require all data underlying the findings described in their manuscript to be freely available to other researchers, either 1. In a public repository, 2. Within the manuscript itself, or 3. Uploaded as supplementary information.

6. Figure 3 includes an image of an identifiable person. Please provide written confirmation or release forms, signed by the subject(s) (or their guardian), giving permission to be photographed and to have their images published under a Creative Commons license. You may upload permission forms to your submission file inventory as item type 'Other'. Otherwise, we kindly request that you remove the photograph.

Additional Editor Comments (if provided):

Reviewers' comments:

Reviewer's Responses to Questions

**Comments to the Author**

1. Does this manuscript meet PLOS Global Public Health’s publication criteria? Is the manuscript technically sound, and do the data support the conclusions? The manuscript must describe methodologically and ethically rigorous research with conclusions that are appropriately drawn based on the data presented.

Reviewer #1: Yes

2. Has the statistical analysis been performed appropriately and rigorously?

Reviewer #1: N/A

3. Have the authors made all data underlying the findings in their manuscript fully available (please refer to the Data Availability Statement at the start of the manuscript PDF file)?

Reviewer #1: Yes

4. Is the manuscript presented in an intelligible fashion and written in standard English?

Reviewer #1: Yes

5. Review Comments to the Author

Reviewer #1: Congratulations on a well written article on an important topic. Some comments for revision

1. The reference to the COVID pandemic does not seem to fit well into the general flow of the manuscript. Please reframe this section.

2. HAPPEE trial could be introduced earlier in the manuscript. When mentioned in line 149, there seems to be no background about what it is and why it is mentioned there.

3. The keywords contains ETHIOPIA which does not seem to be featuring in the manuscript at all

4. The sentence with Ceiling of accountability as mentioned in line 133 is not easily understood. Please reframe.

6. PLOS authors have the option to publish the peer review history of their article (what does this mean?). If published, this will include your full peer review and any attached files.

**Do you want your identity to be public for this peer review?** For information about this choice, including consent withdrawal, please see our Privacy Policy.

Reviewer #1: No

---

## [Decision Letter · Decision Letter 1]

10 Nov 2022

Developing shared understanding of pre-eclampsia in Haiti and Zimbabwe using Theory of Change

PGPH-D-22-00425R1

Dear Dr Robbins,

We are pleased to inform you that your manuscript 'Developing shared understanding of pre-eclampsia in Haiti and Zimbabwe using Theory of Change' has been provisionally accepted for publication in PLOS Global Public Health.

Please ensure that the reference to Fig 3, as mentioned by rev 1 is removed. Please also include some more info in the ethics statement that locals were involved, as per rev 2's request, which the you have already addressed in the Global inclusion questionnaire. Finally, please include that you obtained local ethics approval, which is not in the ethics statement.

Best regards,

Julia Robinson

Executive Editor

Reviewer Comments (if any, and for reference):

Reviewer's Responses to Questions

**Comments to the Author**

1. If the authors have adequately addressed your comments raised in a previous round of review and you feel that this manuscript is now acceptable for publication, you may indicate that here to bypass the “Comments to the Author” section, enter your conflict of interest statement in the “Confidential to Editor” section, and submit your "Accept" recommendation.

Reviewer #1: All comments have been addressed

Reviewer #2: All comments have been addressed

2. Does this manuscript meet PLOS Global Public Health’s publication criteria? Is the manuscript technically sound, and do the data support the conclusions? The manuscript must describe methodologically and ethically rigorous research with conclusions that are appropriately drawn based on the data presented.

Reviewer #1: Yes

Reviewer #2: Yes

3. Has the statistical analysis been performed appropriately and rigorously?

Reviewer #1: N/A

Reviewer #2: N/A

4. Have the authors made all data underlying the findings in their manuscript fully available (please refer to the Data Availability Statement at the start of the manuscript PDF file)?

Reviewer #1: Yes

Reviewer #2: Yes

5. Is the manuscript presented in an intelligible fashion and written in standard English?

Reviewer #1: Yes

Reviewer #2: Yes

6. Review Comments to the Author

Reviewer #1: Thank you for making the required changes. Please also remove reference to Figure 3 in line 208 as the figure has been removed.

Reviewer #2: This is an interesting piece of work and will be a useful addition to the literature. It's difficult to understand without knowing more about the how the study was organised and implemented, but note that there seems to be limited representation of country research team memebers in the authorship list?

7. PLOS authors have the option to publish the peer review history of their article (what does this mean?). If published, this will include your full peer review and any attached files.

**Do you want your identity to be public for this peer review?** For information about this choice, including consent withdrawal, please see our Privacy Policy.

Reviewer #1: No

Reviewer #2: No
